# DppNet: Approximating Determinantal Point Processes with Deep Networks

## Abstract

Determinantal Point Processes (DPPs) provide an elegant and versatile way to sample sets of items that balance the point-wise quality with the set-wise diversity of selected items. For this reason, they have gained prominence in many machine learning applications that rely on subset selection. However, sampling from a DPP over a ground set of size $N$ is a costly operation, requiring in general an $O(N^3)$ preprocessing cost and an $O(Nk^3)$ sampling cost for subsets of size $k$. We approach this problem by introducing DppNets: generative deep models that produce DPP-like samples for arbitrary ground sets. We develop an inhibitive attention mechanism based on transformer networks that captures a notion of dissimilarity between feature vectors. We show theoretically that such an approximation is sensible as it maintains the guarantees of inhibition or dissimilarity that makes DPPs so powerful and unique. Empirically, we demonstrate that samples from our model receive high likelihood under the more expensive DPP alternative.

## 1 Introduction

Selecting a representative sample of data from a large pool of available candidates is an essential step of a large class of machine learning problems: noteworthy examples include automatic summarization, matrix approximation, and minibatch selection. Such problems require sampling schemes that calibrate the tradeoff between the point-wise *quality – e.g.* the relevance of a sentence to a document summary – of selected elements and the set-wise *diversity*[1] of the sampled set as a whole.

Determinantal Point Processes (DPPs) are probabilistic models over subsets of a ground set that elegantly model the tradeoff between these often competing notions of quality and diversity. Given a ground set of size $N$, DPPs allow for $\mathcal{O}(N^3)$ sampling over all $2^N$ possible subsets of elements, assigning to any subset $S$ of a ground set $\mathcal{Y}$ of elements the probability[2]

$$\mathcal{P}_{\boldsymbol{L}}(S) = \det \boldsymbol{L}_S / \det(\boldsymbol{I} + \boldsymbol{L}) \tag{1}$$

where $\boldsymbol{L} \in \mathbb{R}^{N \times N}$ is the DPP kernel and $\boldsymbol{L}_S = [\boldsymbol{L}_{ij}]_{i,j \in S}$ denotes the principal submatrix of $\boldsymbol{L}$ indexed by items in $S$. Intuitively, DPPs measure the volume spanned by the feature embedding of the items in feature space (Figure 1).

First introduced by Macchi (1975) to model the distribution of possible states of fermions obeying the Pauli exclusion principle, the properties of DPPs have since then been studied in depth (Hough et al., 2006; Borodin, 2009, see *e.g.*). As DPPs capture repulsive forces between similar elements, they arise in many natural processes, such as the distribution of non-intersecting random walks (Johansson, 2004), spectra of random matrix ensembles (Mehta & Gaudin, 1960; Ginibre, 1965), and zero-crossings of polynomials with Gaussian coefficients (Hough et al., 2009).

More recently, DPPs have become a prominent tool in machine learning due to their elegance and tractability: recent applications include video recommendation (Chen et al., 2018), minibatch selection (Zhang et al., 2017), and kernel approximation (Li et al., 2016b; Mariet et al., 2018).

However, the $\mathcal{O}(N^3)$ sampling cost makes the practical application of DPPs intractable for large datasets, requiring additional work such as subsampling from $\mathcal{Y}$, structured kernels (Gartrell et al.,

---

[1]Here, we use diversity to mean useful coverage across dissimilar examples in a meaningful feature space, rather than other definitions diversity that may appear in ML fairness literature.

[2]We are adopting here the $\mathcal{L}$-Ensemble construction (Borodin & Rains, 2005) of a DPP.

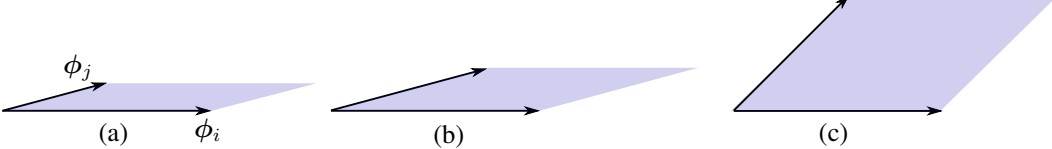

Figure 1: Geometric intuition for DPPs: let $\phi_i, \phi_j$ be two feature vectors of $\mathbf{\Phi}$ such that the DPP kernel verifies $\boldsymbol{L} = \mathbf{\Phi}\mathbf{\Phi}^T$; then $\sqrt{\mathcal{P}_{\boldsymbol{L}}(\{i,j\})} \propto \text{Vol}(\phi_i, \phi_j)$. Increasing the norm of a vector (quality) or increasing the angle between the vectors (diversity) increases the volume spanned by the vectors (Kulesza & Taskar, 2012, Section 2.2.1).

2017; Mariet & Sra, 2016b), or approximate sampling methods (Anari et al., 2016a; Li et al., 2016a; Affandi et al., 2013). Nonetheless, even such methods require significant pre-processing time, and scale poorly with the size of the dataset. Furthermore, when dealing with ground sets with variable components, pre-processing costs cannot be amortized, significantly impeding the application of DPPs in practice.

These setbacks motivate us to investigate the use of more scalable models to generate high-quality, diverse samples from datasets to obtain highly-scalable methods with flexibility to adapt to constantly changing datasets. Specifically, we use generative deep models to approximate the DPP distribution over a ground set of items with both fixed and variable feature representations. We show that a simple, carefully constructed neural network, DPPNET, can generate DPP-like samples with very little overhead, while maintaining fundamental theoretical properties of DPP measures. Furthermore, we show that DPPNETs can be trivially employed to sample from a conditional DPP (i.e. sampling $S$ such that $A \subseteq S$ is predefined) and for greedy mode approximation.

**Contributions.**

- We introduce DPPNET, a deep network trained to generate DPP-like samples based on static and variable ground sets of possible items.

- We derive theoretical conditions under which the DPPNETs inherit the DPP's log-submodularity.

- We show empirically that DPPNETs provide an accurate approximation to DPPs and drastically speed up DPP sampling.

## 2   RELATED WORK

DPPs belong to the class of Strongly Rayleigh (SR) measures; these measures benefit from the strongest characterization of negative association between similar items; as such, SR measures have benefited from significant interest in the mathematics community (Pemantle, 2000; Borcea et al., 2009; Borcea & Brändén, 2008; Marcus et al., 2015) and more recently in machine learning (Anari et al., 2016b; Li et al., 2016a; Mariet et al., 2018). This, combined with their tractability, makes DPPs a particularly attractive tool for subset selection in machine learning, and is one of the key motivations for our work.

The application of DPPs to machine learning problems spans fields from document and video summarization (Gong et al., 2014; Chao et al., 2015), recommender systems (Zhou et al., 2010; Chen et al., 2018) and object retrieval (Affandi et al., 2014) to kernel approximation (Li et al., 2016b), neural network pruning (Mariet & Sra, 2016a), and minibatch selection (Zhang et al., 2017). Zou & Adams (2012) developed DPP priors for encouraging diversity in generative models and Snoek et al. (2013) showed that DPPs accurately model inhibition in neural spiking data.

In the general case, sampling exactly from a DPP requires an initial eigendecomposition of the kernel matrix $\boldsymbol{L}$, incurring a $\mathcal{O}(N^3)$ cost. In order to avoid this time-consuming step, several approximate sampling methods have been derived; Affandi et al. (2013) approximate the DPP kernel during sampling; more recently, results by Anari et al. (2016a) followed by Li et al. (2016a) showed that DPPs are amenable to efficient MCMC-based sampling methods.

Exact methods that significantly speed up sampling by leveraging specific structure in the DPP kernel have also been developed (Mariet & Sra, 2016b; Gartrell et al., 2017). Of particular interest is the dual sampling method introduced in Kulesza & Taskar (2012): if the DPP kernel can be composed as an inner product over a finite basis, i.e. there exists a feature matrix $\Phi \in \mathbb{R}^{N \times D}$ such that the DPP kernel is given by $L = \Phi \Phi^\top$, exact sampling can be done in $\mathcal{O}(ND^2 + NDk^2 + D^2k^3)$.

However, MCMC sampling requires variable amounts of sampling rounds, which is unfavorable for parallelization; dual DPP sampling requires an explicit feature matrix $\Phi$. Motivated by recent work on modeling set functions with neural networks (Zaheer et al., 2017; Cotter et al., 2018), we propose instead to generate approximate samples via a generative network; this allows for simple parallelization while simultaneously benefiting from recent improvements in specialized architectures for neural network models (*e.g.* parallelized matrix multiplications). We furthermore show that, extending the abilities of dual DPP sampling, neural networks may take as input variable feature matrices $\Phi$ and sample from non-linear kernels $L$.

---

**Algorithm 1** Sampling and greedy mode estimation from a DPPNET

1: **function** SAMPLE($k$, $A$)
2:     $S \leftarrow A$
3:     **while** $|S| < k$ **do**
4:         $v \leftarrow \text{DPPNET}(S)$
5:         **if** sampling **then**
6:             $i \sim \text{Multinomial}(v/\|v\|)$
7:         **else if** greedy mode **then**
8:             $i \leftarrow \arg\max v$
9:         $S \leftarrow S \cup \{i\}$
10:     **return** $S$

---

## 3 GENERATING DPP SAMPLES WITH DEEP MODELS

In this section, we build up a framework that allows the $\mathcal{O}(N^3)$ computational cost associated with DPP sampling to be addressed via approximate sampling with a neural network.

Given a positive semi-definite matrix $L \in \mathbb{R}^{N \times N}$, we take $\mathcal{P}_L$ to represent the distribution modeled by a DPP with kernel $L$ over the power set of $[N] := \{1, \ldots, N\}$. Given $A \subseteq [N]$, we write $\bar{A} = [N] \setminus A$ and $L_A = [L_{ij}]_{i,j \in A}$ the $|A| \times |A|$ submatrix of $L$ indexed by items in $A$.

### 3.1 MOTIVATING THE MODEL CHOICE

Although the elegant quality/diversity tradeoff modeled by DPPs is a key reason for their recent success in many different applications, they benefit from other properties that make them particularly well-suited to machine learning problems. We now focus on how these properties can be maintained with a deep generative model with the right architecture.

**Closure under conditioning.** DPPs are closed under conditioning: given $A \subseteq \mathcal{Y}$, the conditional distribution over $\mathcal{Y} \setminus A$ given by $\mathcal{P}_L(S = A \cup B \mid A \subseteq S)$ (for $B \cap A = \emptyset$) is also DPP with kernel $L^A = ([(L + I_{\bar{A}})^{-1}]_{\bar{A}})^{-1} - I$ (Borodin & Rains, 2005). Although conditioning comes at the cost of an expensive matrix inversion, this property make DPPs well-suited to applications requiring diversity in conditioned sets, such as basket completion for recommender systems.

Standard deep generative models such as (Variational) Auto-Encoders (Kingma & Welling, 2014) (VAEs) and Generative Adversarial Networks (Goodfellow et al., 2014) (GANs) would not enable simple conditioning operations during sampling. Instead, we develop a model that given an input set $S$, returns a prediction vector $v \in \mathbb{R}^N$ such that $v_i = \Pr(S \cup \{i\} \subseteq Y \mid S \subseteq Y)$ where $Y \sim \mathcal{P}_L$: in other words, $v_i$ is the marginal probability of item $i$ being included in the final set, given that $S$ is a subset of the final set. Mathematically, we can compute $v_i$ as

$$v_i = 1 - [(L + I_{\bar{S}})^{-1}]_{ii} \tag{2}$$

for $i \notin S$ (Kulesza & Taskar, 2012); for $i \in S$, we simply set $v_i = 0$.

With this architecture, we sample a set via Algorithm 1, which allows for trivial basket-completion type conditioning operations. Furthermore, Algorithm 1 can be modified to implement a greedy sampling algorithm without any additional cost.

**Log-submodularity.** As mentioned above, DPPs are included in the larger class of *Strongly Rayleigh* (SR) measures over subsets. Although being SR is a delicate property, which is maintained

by only few operations (Borcea et al., 2009), log-submodularity[3] (which is implied by SR-ness) is more robust, as well as a fundamental property in discrete optimization (Zagoruyko & Komodakis, 2016; Buchbinder et al., 2012; Grötschel et al., 1981) and machine learning (Iyer & Bilmes, 2015; Wei et al., 2015). Crucially, we show in the following that (log)-submodularity can be inherited by a generative model trained on a log-submodular distribution:

THEOREM 1. *Let $\mathcal{P}$ be a strictly submodular function over subsets of $\mathcal{Y}$, and $\mathcal{Q}$ be a function over the same space such that*

$$D_{\mathrm{TV}}(\mathcal{P}, \mathcal{Q}) \leq \min_{S,T \neq \emptyset, \mathcal{Y}} \frac{1}{4} \left[ \mathcal{P}(S) + \mathcal{P}(T) - \mathcal{P}(S \cup T) - \mathcal{P}(S \cap T) \right] \tag{3}$$

*where $D_{\mathrm{TV}}$ indicates the total variational distance. Then $\mathcal{Q}$ is also submodular.*

COROLLARY 1.1. *Let $\mathcal{P}_{\boldsymbol{L}}$ be a strictly log-submodular DPP over $\mathcal{Y}$ and DPPNET be a network trained on the DPP probabilities $p(S)$, with a loss function of the form $\|\boldsymbol{p} - \boldsymbol{q}\|$ where $\|\cdot\|$ is a norm and $\boldsymbol{p} \in \mathbb{R}^{2^N}$ (resp. $\boldsymbol{q}$) is the probability vector assigned by the DPP (resp. the DPPNET) to each subset of $\mathcal{Y}$. Let $\alpha = \max_{\|\boldsymbol{x}\|_\infty=1} \frac{1}{\|\boldsymbol{x}\|}$. If DPPNET converges to a loss smaller than*

$$\min_{S,T \neq \emptyset, \mathcal{Y}} \frac{1}{4\alpha} \left[ \mathcal{P}(S) + \mathcal{P}(T) - \mathcal{P}(S \cup T) - \mathcal{P}(S \cap T) \right],$$

*its generative distribution is log-submodular.*

The result follows directly from Thm. 3 and the equivalence of norms in finite dimensional spaces.

REMARK 1. Cor. 1.1 is generalizable to the KL divergence loss $D_{\mathrm{KL}}(\mathcal{P}\|\mathcal{Q})$ via Pinsker's inequality.

For this reason, we train our models by minimizing the distance between the predicted and target probabilities, rather than optimizing the log-likelihood of generative samples under the true DPP.

**Leveraging the sampling path.** When drawing samples from a DPP, the standard DPP sampling algorithm (Kulesza & Taskar, 2012, Alg. 1) generates the sample as a sequence, adding items one after the other until reaching a pre-determined size[4], similarly to Alg. 1. We take advantage of this by recording all intermediary subsets generated by the DPP when sampling training data: in practice, instead of training on $n$ subsets of size $k$, we train on $kn$ subsets of size $0, \ldots, k-1$. Thus, our model is very much like an unrolled recurrent neural network.

### 3.2 SAMPLING OVER FIXED AND VARYING GROUND SETS

In the simplest setting, we may wish to draw many samples over a ground set with a fixed feature embedding. In this case, we wish to model a DPP with a fixed kernel via a generative neural network. Specifically, we consider a fixed DPP with kernel $\boldsymbol{L}$ and wish to obtain a generative model such that its distribution $\mathcal{G} : 2^{\mathcal{Y}} \to [0,1]$ is close (under a chosen metric) to the DPP distribution $\mathcal{P}_{\boldsymbol{L}}$.

More generally, in many cases we may care about sampling from a DPP over a ground set of items that varies: this may be the case for example with a pool of products that are available for sale at a given time, or social media posts with a relevance that varies based on context. To leverage the speed-up provided by dual DPP sampling, we can only sample from the DPP with kernel given by $\boldsymbol{L} = \boldsymbol{\Phi}\boldsymbol{\Phi}^\top$; for more complex kernels, we once again incur the $\mathcal{O}(N^3)$ cost. Furthermore, training a static neural network for each new feature embedding may be too costly.

Instead, we augment the static DPPNET to include the feature matrix $\boldsymbol{\Phi}$ representing the ground set of all items as input to the network. Specifically, we draw inspiration for the dot-product attention introduced in (Vaswani et al., 2017). In the original paper, the attention mechanism takes as input 3 matrices: the keys $\boldsymbol{K}$, the values $\boldsymbol{V}$, and the query $\boldsymbol{Q}$. Attention is computed as $\boldsymbol{A} = \mathrm{softmax}(\boldsymbol{Q}\boldsymbol{K}^\top/\sqrt{d})$ where $d$ is the dimension of each query/key: the inner product acts as a proxy to the similarity between each query and each key. Finally, the reweighted value matrix $\boldsymbol{A}\boldsymbol{V}$ is fed to the trainable neural network.

---

[3]Recall that a function $f$ over subsets is log-submodular if and only if for any sets $S, T$, we have $\log f(S) + \log f(T) - \log f(S \cup T) - \log f(S \cap T) \geq 0$.

[4]When not fixed by the user as in $k$-DPPs (Kulesza & Taskar, 2011), the expected sampled set size under a DPP depends on the eigenspectrum of the kernel $\boldsymbol{L}$.

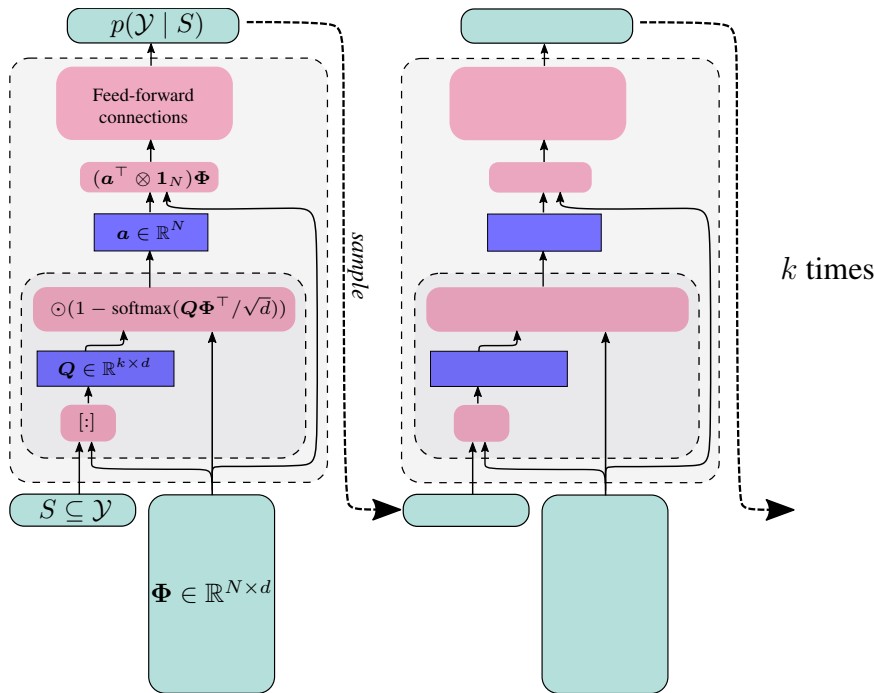

Figure 2: Transformer network architecture for sampling from a variable ground set.

Here, the feature representation of items in the input set $S$ acts as the query $\boldsymbol{Q} \in \mathbb{R}^{k \times d}$; the feature representation $\boldsymbol{\Phi} \in \mathbb{R}^{N \times d}$ of our ground set is both the keys and the values. In order for the attention mechanism to make sense in the framework of DPP modeling, we make two modifications to the attention in (Vaswani et al., 2017):

- We want our network to attend to items that are *dissimilar* to the query (input subset): for each item $i$ in the input subset $S$, we compute its pairwise dissimilarity to each item in $\mathcal{Y}$ as the vector $\boldsymbol{d}_i = 1 - \mathrm{softmax}(\boldsymbol{Q}_i \boldsymbol{\Phi}^\top / \sqrt{d})$.

- Instead of returning this $k \times N$ matrix $\boldsymbol{D}$ of dissimilarities $\boldsymbol{d}_i$, we return a vector $\boldsymbol{a} \in \mathbb{R}^N$ in the probability simplex such that $a_j \propto \prod_{i \in S} \boldsymbol{D}_{ij}$. This allows us to have a fixed-size input to the neural network, and simultaneously enforces the desirable property that similarity to a *single* item is enough to disqualify an element from the ground set. Note that we could also return $\boldsymbol{D}$ in the form of a $N \times N$ matrix, but this would be counterproductive to speeding up DPP sampling.

Putting everything together, our attention vector $\boldsymbol{a}$ is computed via the *inhibitive attention* mechanism

$$\boldsymbol{a}' = \underset{i \in S}{\odot} \left( 1 - \mathrm{softmax}(\boldsymbol{Q}\boldsymbol{\Phi}^\top / \sqrt{d}) \right), \qquad \boldsymbol{a} = \boldsymbol{a}' / \|\boldsymbol{a}'\|_1 \tag{4}$$

where $\odot$ represents the row-wise multiplication operator; this vector can be computed in $\mathcal{O}(kDN)$ time. The attention component of the neural network finally feeds the element-wise multiplication of each row of $\boldsymbol{V}$ with $\boldsymbol{a}$ to the feed-forward component.

Given $\boldsymbol{\Phi}$ and a subset $S$, the network is trained as in the static case to learn the marginal probabilities of adding any item in $\mathcal{Y}$ to $S$ under a DPP with a kernel $\boldsymbol{L}$ dependent on $\boldsymbol{\Phi}$. In practice, we set $\boldsymbol{L}$ to be an exponentiated quadratic kernel $\boldsymbol{L}_{ij} = \exp(-\beta \|\phi_i - \phi_j\|^2)$ constructed with the features $\phi_i$.

REMARK 2. Dual sampling for DPPs as introduced in (Kulesza & Taskar, 2012) is efficient only when sampling from a DPP with kernel $\boldsymbol{L} = \boldsymbol{\Phi}\boldsymbol{\Phi}^\top$; for non-linear kernels, a low-rank decomposition of $\boldsymbol{L}(\boldsymbol{\Phi})$ must first be obtained, which in the worst case requires $\mathcal{O}(N^3)$ operations. In comparison, the dynamic DPPNET can be trained on any DPP kernel, while only requiring $\boldsymbol{\Phi}$ as input.

Table 1: NLL under $\mathcal{P}_L$ for sets of size $k = 20$ sampled over the unit square. DPPNET achieves comparable performance to the DPP, outperforming the other baselines.

| UNIFORM | $k$-MEDOIDS | DPPNET | DPP |
|---|---|---|---|
| $180.53 \pm 9.56$ | $169.37 \pm 6.41$ | $153.44 \pm 2.07$ | $154.95 \pm 2.93$ |

## 4 EXPERIMENTAL RESULTS

To evaluate DPPNET, we look at its performance both as a proxy for a static DPP (Section 4.1) and as a tool for generating diverse subsets of varying ground sets (Section 4.2). Our models are trained with TensorFlow, using the Adam optimizer. Hyperparameters are tuned to maximize the normalized log-likelihood of generated subsets.

We compare DPPNET to DPP performance as well as two additional baselines:

- UNIF: Uniform sampling over the ground set.
- $k$-MEDOIDS: The $k$-medoids clustering algorithm (Hastie et al., 2001, 14.3.10), applied to items in the ground set, with distance between points computed as the same distance metric used by the DPP. Conversely to $k$-means, $k$-MED uses data points as centers for each cluster.

We use the negative log-likelihood (NLL) of a subset under a DPP constructed over the ground set to evaluate the subsets obtained by all methods. This choice is motivated by the following considerations: DPPs have become a standard way of measuring and enforcing diversity over subsets of data in machine learning, and b) to the extent of our knowledge, there is no other standard method to benchmark the diversity of a selected subset that depends on specific dataset encodings.

### 4.1 SAMPLING OVER THE UNIT SQUARE

We begin by analyzing the performance of a DPPNET trained on a DPP with fixed kernel over the unit square. This is motivated by the need for diverse sampling methods on the unit hypercube, motivated by *e.g.* quasi-Monte Carlo methods, latin hypercube sampling (McKay et al., 1979) and low discrepancy sequences.

The ground set consists of the 100 points lying at the intersections of the $10 \times 10$ grid on the unit square. The DPP is defined by setting its kernel $L$ to $L_{ij} = \exp(-\|x_i - x_j\|_2^2/2)$. As the DPP kernel is fixed, these experiments exclude the inhibitive attention mechanism.

We report the performance of the different sampling methods in Figure 3. Visually (Figure 3a) and quantitively (Figure 3b), DPPNET improves significantly over all other baselines. The NLL of DPPNET samples is almost identical to that of true DPP samples. Furthermore, greedily sampling the mode from the DPPNET achieves a better NLL than DPP samples themselves. Numerical results are reported in Table 1.

### 4.2 SAMPLING OVER VARIABLE GROUND SETS

We evaluate the performance of DPPNETs on varying ground set sizes through the MNIST (LeCun & Cortes, 2010), CelebA (Liu et al., 2015), and MovieLens (Harper & Konstan, 2015) datasets. For MNIST and CelebA, we generate feature representations of length 32 by training a Variational Auto-Encoder (Kingma & Welling, 2014) on the dataset[5]; for MovieLens, we obtain a feature vector for each movie by applying nonnegative matrix factorization the rating matrix, obtaining features of length 10. Experimental results presented below were obtained using feature representations obtained via the test instances of each dataset.

The DPPNET is trained based on samples from DPPs with a linear kernel for MovieLens and with an exponentiated quadratic kernel for the image datasets. Bandwidths were set to $\beta = 0.0025$ for MNIST and $\beta = 0.1$ for CelebA in order to obtain a DPP average sample size $\approx 20$: recall that for a DPP with kernel $L$, the expected sample size is given by the formula

$$\mathbb{E}_{S \sim \mathcal{P}_L}[|S|] = \text{Tr}[L(L + I)^{-1}].$$

---

[5]Details on the encoders are provided in Appendix B.

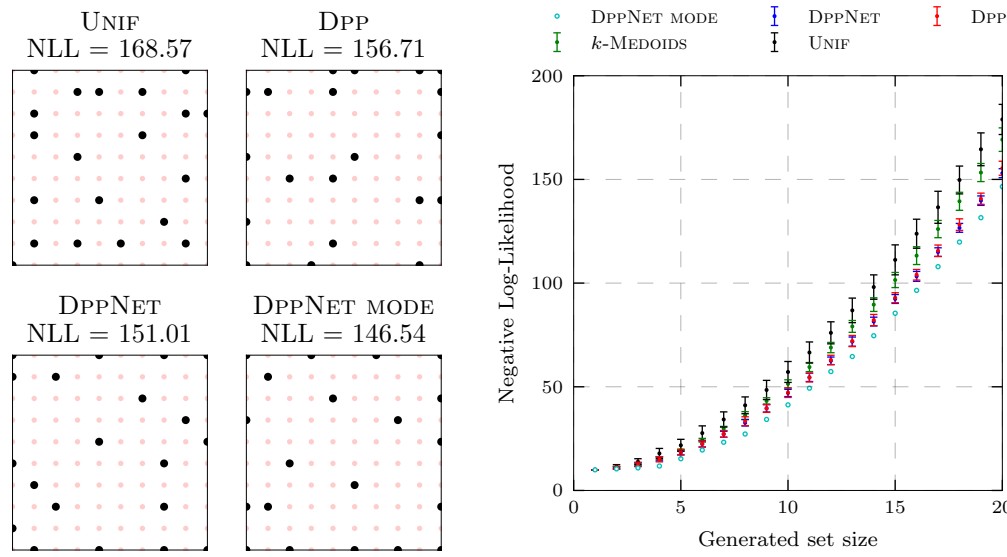

(a) Sampling subsets of size 20 over the unit square.

(b) Normalized log-likelihood of samples drawn from all methods as a function of the sampled set size.

Figure 3: Sampling on the unit square with a DPPNET (1 hidden layer with 841 neurons) trained on a single DPP kernel. Visually, DPPNET gives similar results to the full DPP (left). As evaluated by DPP NLL, the DPPNET's mode achieves superior performance to the full DPP, and DPPNET sampling overlaps completely with DPP sampling (right).

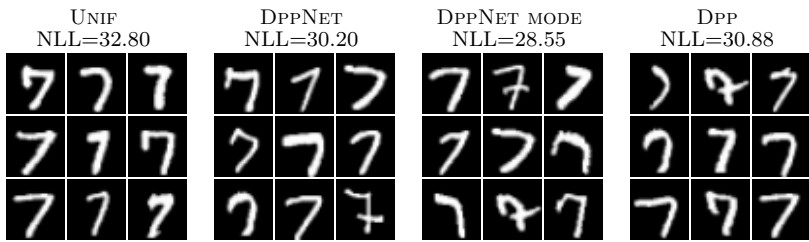

Figure 4: Digits sampled from a DPPNET (3 layer of 365 neurons) trained on MNIST encodings.

For MNIST, Figure 4 shows images selected by the baselines and the DPPNET, chosen among 100 digits with either random labels or all identical labels; visually, DPPNET and DPP samples provide a wider coverage of writing styles. However, the NLL of samples from DPPNET decay significantly, whereas the DPPNET mode continues to maintain competitive performance with DPP samples.

Numerical results for MNIST are reported in Table 2; additionally to the previous baselines, we also consider two further ways of generating subsets. INHIBATTN samples items from the multinomial distribution generated by the inhibitive attention mechanism only (without the subsequent neural network). NOATTN is a pure feed-forward neural network without attention; after hyper-parameter tuning, we found that the best architecture for this model consisted in 6 layers of 585 neurons each.

Table 2 reveals that both the attention mechanism and the subsequent neural network are crucial to modeling DPP samples. Strikingly, DPPNET performs significantly better than other baselines even on feature matrices drawn from a single class of digits (Table 2), despite the training distribution over feature matrices being much less specialized. This implies that DPPNET sampling for dataset summarization may be leveraged to focus on sub-areas of datasets that are identified as areas of interest. Numerical results for CelebA and MovieLens are reported in Table 3, confirming the modeling ability of DPPNETs.

Finally, we verify that DPPNET allows for significantly faster sampling by running DPP and DPPNET sampling for subsets of size 20 drawn from a ground set of size 100 with both a standard DPP

Table 2: NLL (mean $\pm$ standard error) under the true DPP of samples drawn uniformly, according to the mode of the DPPNET, and from the DPP itself. We sample subsets of size 20; for each class of digits we build 25 feature matrices $\Phi$ from encodings of those digits, and for each feature matrix we draw 25 different samples. Bolded numbers indicate the best-performing (non-DPP) sampling method.

| | All | 0 | 1 | 2 | 3 | 4 | 5 | 6 | 7 | 8 | 9 |
|---|---|---|---|---|---|---|---|---|---|---|---|
| DPP | $49.2 \pm 0.1$ | $\mathbf{52.2 \pm 0.1}$ | $60.5 \pm 0.1$ | $49.8 \pm 0.0$ | $50.7 \pm 0.1$ | $51.0 \pm 0.1$ | $50.4 \pm 0.1$ | $51.6 \pm 0.1$ | $51.5 \pm 0.1$ | $50.9 \pm 0.1$ | $52.7 \pm 0.1$ |
| UNIF | $51.6 \pm 0.1$ | $54.9 \pm 0.1$ | $65.1 \pm 0.1$ | $51.5 \pm 0.1$ | $52.9 \pm 0.1$ | $53.3 \pm 0.1$ | $52.4 \pm 0.1$ | $54.6 \pm 0.1$ | $55.1 \pm 0.1$ | $53.3 \pm 0.1$ | $56.2 \pm 0.1$ |
| MEDOIDS | $51.0 \pm 0.1$ | $55.1 \pm 0.1$ | $65.0 \pm 0.1$ | $51.5 \pm 0.1$ | $52.9 \pm 0.1$ | $53.1 \pm 0.1$ | $52.4 \pm 0.0$ | $54.4 \pm 0.1$ | $55.1 \pm 0.1$ | $53.2 \pm 0.1$ | $56.1 \pm 0.1$ |
| INHIBATTN | $51.3 \pm 0.1$ | $54.7 \pm 0.1$ | $65.0 \pm 0.1$ | $51.4 \pm 0.1$ | $52.8 \pm 0.1$ | $53.0 \pm 0.1$ | $52.1 \pm 0.1$ | $54.5 \pm 0.1$ | $54.9 \pm 0.1$ | $53.2 \pm 0.1$ | $55.9 \pm 0.1$ |
| NOATTN | $51.4 \pm 0.1$ | $54.9 \pm 0.1$ | $65.4 \pm 0.1$ | $51.5 \pm 0.1$ | $52.9 \pm 0.1$ | $53.3 \pm 0.1$ | $52.2 \pm 0.1$ | $54.6 \pm 0.1$ | $55.2 \pm 0.1$ | $53.3 \pm 0.1$ | $56.1 \pm 0.1$ |
| DPPNET MODE | $48.6 \pm 0.2$ | $\mathbf{53.6 \pm 0.3}$ | $\mathbf{63.6 \pm 0.4}$ | $\mathbf{50.8 \pm 0.2}$ | $\mathbf{51.4 \pm 0.3}$ | $\mathbf{51.6 \pm 0.4}$ | $\mathbf{51.8 \pm 0.3}$ | $\mathbf{52.8 \pm 0.3}$ | $\mathbf{52.7 \pm 0.4}$ | $\mathbf{50.9 \pm 0.3}$ | $\mathbf{55.0 \pm 0.4}$ |

Table 3: NLLs on CelebA and MovieLens (mean $\pm$ standard error); 20 samples of size 20 were drawn across 20 different feature matrices each for a total of 100 samples per method; DPPNET is the non-DPP model achieves the best NLLs.

| DATASET | KERNEL | DPP GOAL | UNIFORM | $k$-MEDOIDS | DPPNET Mode |
|---|---|---|---|---|---|
| CelebA | Exp. quadratic | $49.04 \pm 2.03$ | $50.84 \pm 1.53$ | $51.18 \pm 1.34$ | $\mathbf{49.28 \pm 1.57}$ |
| MovieLens | Linear | $84.29 \pm 0.20$ | $92.04 \pm 0.17$ | $88.90 \pm 0.16$ | $\mathbf{80.21 \pm 0.33}$ |

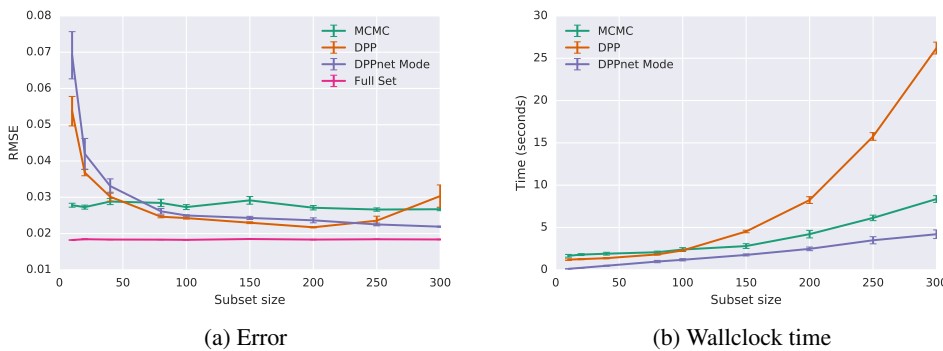

(a) Error

(b) Wallclock time

Figure 5: Results for the Nyström approximation experiments, comparing DPPNET to the fast MCMC sampling method of Li et al. (2016c) according to root mean squared error (RMSE) and wallclock time. We observe that subsets selected by DPPNET achieve comparable and lower RMSE than a DPP and the MCMC method respectively while being significantly faster.

and DPPNET (using the MNIST architecture). Both methods were implemented in graph-mode TensorFlow. Sampling batches of size 32, standard DPP sampling costs $2.74 \pm 0.02$ seconds; DPPNET sampling takes $0.10 \pm 0.001$ seconds, amounting to an almost 30-fold speed improvement.

### 4.3 DPPNET FOR KERNEL RECONSTRUCTION

As a final experiment, we evaluate DPPNET's performance on a downstream task for which DPPs have been shown to be useful: kernel reconstruction using the Nyström method (Nyström, 1930; Williams & Seeger, 2001). Given a positive semidefinite matrix $\mathbf{K} \in \mathbb{R}^{N \times N}$, the Nyström method approximates $\mathbf{K}$ by $\hat{\mathbf{K}} = \mathbf{K}_{.,S} \mathbf{K}_{S,S}^{\dagger} \mathbf{K}_{S,.}$ where $\mathbf{K}^{\dagger}$ denotes the pseudoinverse of $\mathbf{K}$ and $\mathbf{K}_{.,S}$ (resp. $\mathbf{K}_{S,.}$) is the submatrix of $\mathbf{K}$ formed by its rows (resp. columns) indexed by $S$. The Nyström method is a popular method to scale up kernel methods and has found many applications in machine learning (see *e.g.* (Bac; She; Fow; Tal)). Importantly, the approximation quality directly depends on the choice of subset $S$. Recently, DPPs have been shown to be a competitive approach for selecting $S$ (Mariet et al., 2018; Li et al., 2016b). Following the approach of Li et al. (2016b), we evaluate the quality of the kernel reconstruction by learning a regression kernel $\mathbf{K}$ on a training set, and reporting the prediction error on the test set using the Nyström reconstructed kernel $\hat{\mathbf{K}}$. Additionally to the full DPP, we also compare DPPNET to the MCMC sampling method with quadrature acceleration (Li

et al., 2016b;c) Figure 5 reports our results on the Ailerons dataset[6] also used in Li et al. (2016b). We start with a ground set size of 1000 and compute the resulting root mean squared error (RMSE) of the regression using various sized subsets selected by sampling from a DPP, the MCMC method of Li et al. (2016c), using the full ground set and DPPNET. Figure 5b reports the runtimes for each method. We note that while all methods were run on CPU, DPPNet is more amenable to acceleration using GPUs.

## 5 Conclusion and discussion

We introduced DPPNETs, generative networks trained on DPPs over static and varying ground sets which enable fast and modular sampling in a wide variety of scenarios. We showed experimentally on several datasets and standard DPP applications that DPPNETs obtain competitive performance as evaluated in terms of NLLs, while being amenable to the extensive recent advances in speeding up computation for neural network architectures.

Although we trained our models on DPPs on exponentiated quadratic and linear kernels; we can train on any kernel type built from a feature representations of the dataset. This is not the case for dual DPP exact sampling, which requires that the DPP kernel be $L = \Phi\Phi^\top$ for faster sampling.

DPPNETs are *not* exchangeable: that is, two sequences $i_1, \ldots, i_k$ and $\sigma(i_1), \ldots, \sigma(i_k)$ where $\sigma$ is a permutation of $[k]$, which represent the same set of items, will not in general have the same probability under a DPPNET. Exchangeability can be enforced by leveraging previous work (Zaheer et al., 2017); however, non-exchangeability can be an asset when sampling a ranking of items.

Our models are trained to take as input a fixed-size subset representation; we aim to investigate the ability to take a variable-length encoding as input as future work. The scaling of the DPPNET's complexity with the ground set size also remains an open question. However, standard tricks to enforce fixed-size ground sets such as sub-sampling from the dataset may be applied to DPPNETs. Similarly, if further speedups are necessary, sub-sampling from the ground set – a standard approach for DPP sampling over very large set sizes – can be combined with DPPNET sampling.

In light of our results on dataset sampling, the question of whether encoders can be trained to produce encodings conducive to dataset summarization via DPPNETs seems of particular interest. Assuming knowledge of the (encoding-independent) relative diversity of a large quantity of subsets, an end-to-end training of the encoder and the DPPNET simultaneously may yield interesting results.

Finally, although Corollary 1.1 shows the log-submodularity of the DPP can be transferred to a generative model, understanding which additional properties of training distributions may be conserved through careful training remains an open question which we believe to be of high significance to the machine learning community in general.

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

## A    MAINTAINING LOG-SUBMODULARITY IN THE GENERATIVE MODEL

THEOREM 2. *Let $p$ be a strictly submodular distribution over subsets of a ground set $\mathcal{Y}$, and $q$ be a distribution over the same space such that*

$$D_{TV}(p,q) \leq \min_{S,T \neq \emptyset, \mathcal{Y}} \frac{1}{4} \left[ p(S) + p(T) - p(S \cup T) - p(S \cap T)) \right]. \tag{5}$$

*Then $q$ is also submodular.*

*Proof.*  In all the following, we assume that $S, T$ are subsets of a ground set $\mathcal{Y}$ such that $S \neq T$ and $S, T \notin \{\emptyset, \mathcal{Y}\}$ (the inequalities being immediate in these corner cases).

Let
$$\epsilon := \min_{S,T} p(S) + p(T) - p(S \cup T) - p(S \cap T))$$

By the strict submodularity hypothesis, we know $\epsilon > 0$.

Let $S, T \subseteq \mathcal{Y}$ such that $S \neq T$ and $S, T \neq \emptyset, \mathcal{Y}$. To show the log-submodularity of $q$, it suffices to show that
$$q(S) + q(T) \geq q(S \cup T) + q(S \cap T).$$

By definition of $\epsilon$,
$$p(S) + p(T) - p(S \cup T) - p(S \cap T)) \geq \epsilon$$

From equation 5, we know that
$$\max_{S \subseteq \mathcal{Y}} |p(S) - q(S)| \leq \epsilon/4.$$

It follows that
$$q(S) + q(T) - q(S \cup T) + q(S \cap T) \geq p(S) + p(T) - p(S \cup T) - p(S \cap T) - \epsilon$$
$$\geq 0$$

which proves the submodularity of $q$. $\qquad\qquad\square$

## B    ENCODER DETAILS

For the MNIST encodings, the VAE encoder consists of a 2d-convolutional layer with 64 filters of height and width 4 and strides of 2, followed by a 2d convolution layer with 128 filters (same height, width and strides), then by a dense layer of 1024 neurons. The encodings are of length 32.

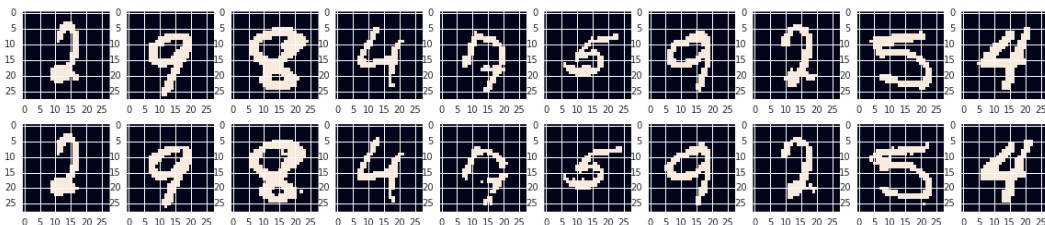

Figure 6: Digits and VAE reconstructions from the MNIST training set

CelebA encodings were generated by a VAE using a Wide Residual Network (Zagoruyko & Komodakis, 2016) encoder with 10 layers and filter-multiplier $k = 4$, a latent space of 32 full-covariance Gaussians, and a deconvolutional decoder trained end-to-end using an ELBO loss. In detail, the decoder architecture consists of a 16K dense layer followed by a sequence of $4 \times 4$ convolutions with [512, 256, 128, 64] filters interleaved with $2\times$ upsampling layers and a final $6 \times 6$ convolution with 3 output channels for each of 5 components in a mixture of quantized logistic distributions representing the decoded image.

