# OpenReview forum: "DppNet: Approximating Determinantal Point Processes with Deep Networks"
_ICLR.cc/2019/Conference_

### Official Review · AnonReviewer3 · 2018-11-02
**This paper proposes DppNet, which approximates determinantal point processes with deep networks by inhibitive attention mechanism. The authors provided a theoretical analysis under some condition that the DppNet is of log-submodularity. Further, some experiments are conducted to show the performance.**

**Rating:** 5
**Confidence:** 3

**Review:**

Quality (5/10): This paper proposes DppNet, which approximates determinantal point processes with deep networks by inhibitive attention mechanism. The authors provided a theoretical analysis under some condition that the DppNet is of log-submodularity.

Clarity (9/10): This paper is well written and provides a clear figure to demonstrate their network architecture.

Originality (6/10): This paper is mainly based on the work [Vaswani et al, Attention is all you need, 2017]. It computes the dissimilarities by subtracting attention in the original work from one, and then samples a subset by an unrolled recurrent neural network.

Significance (5/10): This paper uses negative log-likelihood as the measurement to compare DppNet with other methods. Without further application, it is difficult to measure the improvement of this method over other methods.

Pros:
(1) This paper is well written and provides a figure to clearly demonstrate their network architecture.

(2) This paper provides a deep learning way to sample a subset of data from the whole data set and reduce the computation complexity.

There are some comments.
(1) Figure 4 shows the sampled digits from Uniform distribution, DppNet (with Mode) and Dpp. How about the sampled digits from k-Medoids? Providing the sampled digits from k-Medoids can make the experiments more complete.

(2) The object of DppNet is to minimize the negative log-likelihood. The DPP and k-Medoids have other motivations, not directly optimizing the negative log-likelihood. This may be the reason why DppNet has a better performance on negative log-likelihood, even than DPP. Could the authors provide some other measures (like the visual comparison in figure 4) to compare these methods?

(3) Does GenDpp Mode in Table 2 mean the greedy mode in Algorithm 1? A clear denotation can make it more clear.

---

> ### Author Response · Authors · 2018-11-26
> **Clarifications**
>
> Thank you for your comments; we hope the below clarifications answer your questions.
>
> (1) Sampled digits from k-medoids: Thank you for catching this oversight; we will include the k-medoid samples in the updated paper.
> (2) Other measure of performance: we will add an additional evaluation on a downstream task, using DppNet and other baseline methods to sample columns to reconstruct a large kernel with the Nystrom method.
> (3) GenDpp mode: Yes, this indicates the greedy mode of Algorithm 1; we will clarify this.

---

### Official Review · AnonReviewer2 · 2018-11-05
**Interesting paper with good ideas but limited applicability (in its current form)**

**Rating:** 5
**Confidence:** 4

**Review:**

This paper proposes a scaleable algorithm for sampling from DppNets, a proposed model which approximates the distribution of a DPP. The approach builds upon a proposed inhibitive attention mechanism and transformer networks.

The proposed approach and focus on sampling is original as far as I can tell. The problem is also important to parts of the community as DPPs (or similar distributions) are used more and more frequently. However, the applicability of the proposed approach is limited as it is unclear how to deal with varying ground set sizes — the authors briefly discuss this issue in their conclusion referring to circumvent this problem by subsampling (this can however be problematic either requiring to sample from a DPP or incurring high probability of missing „important“ items).

Furthermore the used evaluation method is „biased“ in favor of DppNets as numerical results evaluate the likelihood of samples under the DPP which the DppNet is trained to approximate for. This makes it difficult to draw conclusions from the presented results. I understand that this evaluation is used as there is no standard way of measuring diversity of a subset of items, but it is also clear that „no“ baseline can be competitive. One possibility to overcome this bias would be to consider a downstream task and evaluate performance on that task.

Furthermore, I suggest to make certain aspects of the paper more explicit and provide additional details. For instance, I would suggest to spell out a training algorithm, provide equations for the training criterion and the evaluation criterion. Please comment on the cost of training (constantly computing the marginal probabilities for training should be quite expensive) and the convergence of the training (maybe show a training curve; this would be interesting in the light of Theorem 1 and Corollary 1).

Certain parts of the paper are unclear or details are missing:
* Table 3: What is „DPP Gao“?
* How are results for k-medoids computed (including the standard error)? Are these results obtained by computing multiple k-medoids solutions with differing initial conditions?
* In the paper you say: „Furthermore, greedily sampling the mode from the DPPNET achieves a better NLL than DPP samples themselves.“ What are the implications of this? What is the NLL of an (approximate) mode of the original DPP? Is the statement that you want to make, that the greedy approximation works well?

---

> ### Author Response · Authors · 2018-11-26
> **Several clarifications**
>
> Thank you for your suggestions regarding the clarity of the paper; we will augment our work with all suggested algorithms and equations.
>
> Limited applicability to variable ground set sizes: this is a drawback of our current approach. However, one can easily circumvent this problem in cases where an upper bound N_max on ground set sizes is known: train a DppNet with ground set size N_max, and in all cases where N <= N_max, complete the missing items with placeholder 0 vectors. Algorithm 1 can be trivially modified to take this into account to ensure that these dummy items are not selected.
>
> Evaluation biased towards DppNets: our goal is to show that DppNet approximates DPP-like samples (much) better than other reasonable approximations. We did not originally include downstream tasks as DPP have been accepted as a state-of-the-art method for diverse sampling in ML applications (see e.g. recent work such as https://dl.acm.org/citation.cfm?id=3272018 for real-world applications). However, as mentionned to Reviewer 1, we will include a downstream task (kernel reconstruction via the Nystrom method) to further support our claims.
>
> Cost of evaluating the marginal probabilities: indeed, this is the costly part of our algorithm (which only impacts training time); this cost is mitigated by two aspects:
> - Given S, we can compute the probability P(S U {i} | S) for all i simultaneously with no overhead (Eq. 2)
> - When training a DppNet over varying ground sets, this cost is offset by the fact that we are not learning one but a whole class of DPPs simultaneously.
>
> * “DPP Gao”: This is a typo; it should read “DPP Goal”
> * K-medoids: Yes, we run the algorithm multiple times with different initializations.
> * Greedy sampling: Yes, we are stating that greedy sampling with DppNet yields realistic DPP samples, as evidenced by the high DPP log-likelihoods. This is a significant advantage over standard DPPs, since the greedy DPP mode algorithm is costly even with recent improvements [NIPS ‘18, Hulu].

---

### Official Review · AnonReviewer1 · 2018-11-05
**comparison with faster algorithms for sampling from DPPs**

**Rating:** 3
**Confidence:** 5

**Review:**

Determinantal Point Processes provide an efficient and elegant way to sample a subset of diverse items from a ground set. This has found applications in summarization, matrix approximation, minibatch selection. However, the naive algorithm for DPP takes time O(N^3), where N is the size of the ground set. The authors provide an alternative model DPPNet for sampling diverse items that preserves the elegant mathematical properties (closure under conditioning, log-submodularity) of DPPs while having faster sampling algorithms.

The authors need to compare the performance of DPPNet against faster alternatives to sample from DPPs, e.g., https://arxiv.org/pdf/1509.01618.pdf, as well as compare on applications where there is a significant gap between uniform sampling and DPPs (because there are the applications where DPPs are crucial). The examples in Table 2 and Table 3 do not address this.

---

> ### Author Response · Authors · 2018-11-26
> **Expected runtimes of various DPP sampling methods; only DppNet benefits from hardware acceleration.**
>
> Thank you for your feedback. We will include a comparison to other approximate sampling methods such as the coresets method you mentioned in an updated version of the paper.
>
> However, we would like to insist upon the following:
> - The runtime of the coreset sampling method is O(Nk^3), which is the same as the runtime as dual DPP sampling discussed in section 2.
> - The coreset approach does not have a hardware accelerator-friendly implementation, as it requires iteratively computing elementary symmetric polynomials and many sequential operations; the same holds of MCMC sampling methods.
> For this reason, we expect to see DppNet have a drastically faster runtime even when compared to such methods on small datasets.
>
> Regarding evaluating DppNet to other methods on applications where there is a gap between uniform and DPP sampling, we agree that doing so will increase the impact of our paper. We are planning on augmenting our experimental section with an evaluation of all methods on the task of reconstructing large kernels via the Nystrom method.

---

### Author Response · Authors · 2018-11-25
**Clarification re: experimental section**

We thank the reviewers for their detailed comments. We would like to clarify the aim of our experimental section: over the past years DPPs have been proven crucial to modeling diversity and quality trade-offs in subset selection problems (recommender systems, kernel reconstruction, …). For this reason, our experiments aim to show that DppNets approximate DPPs better than other reasonable baselines (which we show by comparing NLLs under the true DPP). Crucially, our experiments do not aim to show that DppNet generates more diverse subsets: showing that DppNet is close to DPP samples is sufficient.

However, based on the feedback we have received, we plan on incorporating additional experiments to an updated version of the paper to show that DppNet’s DPP-like samples imply good performance on downstream tasks where DPPs have been shown to be valuable.

---

> ### Author Response · Authors · 2018-11-27
> **Additional experiments have been included**
>
> Following the recommendation of the reviewers, we have added two experiments to our paper:
>
> - A timing comparison between MCMC sampling and DppNet, which shows that DppNet is significantly faster than MCMC sampling.
> - An evaluation of DppNet for kernel reconstruction via the Nystrom method (a downstream task for which DPPs are known to be successful), which we compare to standard and MCMC sampling. In practice, we see that DppNet's performance on this task matches or outperforms the other baselines.
>
> Put together, these experiments show that DppNet is significantly faster than MCMC, while being competitive (or outperforming) MCMC on downstream DPP tasks.

---

### Meta-Review · Area_Chair1 · 2018-12-12
**Limited applicability**

**Confidence:** 4
**Recommendation:** Reject

**Metareview:**

The paper addresses the complexity issue of Determinantal Point Processes via generative deep models.

The reviewers and AC note the critical limitation of applicability of this paper to variable ground set sizes, whether authors' rebuttal is not convincing enough.

AC thinks the proposed method has potential and is interesting, but decided that the authors need more works to publish.